# Simulation of a Composite with a Polyhydroxybutyrate (PHB) Matrix Reinforced with Cylindrical Inclusions: Prediction of Mechanical Properties

**DOI:** 10.3390/polym15244727

**Published:** 2023-12-17

**Authors:** Natalia Gómez-Gast, Juan Andrés Rivera-Santana, José A. Otero, Horacio Vieyra

**Affiliations:** 1Tecnologico de Monterrey, Escuela de Ingeniería y Ciencias, Carretera Lago de Guadalupe 3.5, Colonia Margarita Maza de Juárez, Atizapán de Zaragoza 52926, Mexico or nataliagast@gmail.com (N.G.-G.); j.a.otero@tec.mx (J.A.O.); 2Tecnologico de Monterrey, Escuela de Ingeniería y Ciencias, Eduardo Monroy Cárdenas 2000, San Antonio Buenavista, Toluca de Lerdo 50110, Mexico; 3Escuela de Ingeniería, Cetys Universidad, Campus Mexicali, Calzada Cetys, s/n, Colonia Rivera, Mexicali 21259, Mexico; juanandres.rivera@cetys.mx

**Keywords:** simulation, polymer properties, finite element, biocomposites, biodegradable matrix, cylindrical inclusions

## Abstract

Biocomposite development, as a sustainable alternative to fossil-derived materials with diverse industrial applications, requires expediting the design process and reducing production costs. Simulation methods offer a solution to these challenges. The main aspects to consider in simulating composite materials successfully include accurately representing microstructure geometry, carefully selecting mesh elements, establishing appropriate boundary conditions representing system forces, utilizing an efficient numerical method to accelerate simulations, and incorporating statistical tools like experimental designs and re-regression models. This study proposes a comprehensive methodology encompassing these aspects. We present the simulation using a numerical homogenization technique based on FEM to analyze the mechanical behavior of a composite material of a polyhydroxybutyrate (PHB) biodegradable matrix reinforced with cylindrical inclusions of flax and kenab. Here, the representative volume element (RVE) considered the geometry, and the numerical homogenization method (NHM) calculated the macro-mechanical behavior of composites. The results were validated using the asymptotic homogenization method (AHM) and experimental data, with error estimations of 0.0019% and 7%, respectively. This model is valuable for predicting longitudinal and transverse elastic moduli, shear modulus, and Poisson’s coefficient, emphasizing its significance in composite materials research.

## 1. Introduction

Biocomposites are mixtures of at least two phases, a matrix and a reinforcement, with at least one biodegradable component [1,2,3]. The matrix may be biodegradable, non-biodegradable, or partially bioderived [4,5,6,7,8,9]. The reinforcements can be naturally derived from animals, minerals, and plants or be synthetic [10,11,12]. When the matrix is not biodegradable but the reinforcement is, it is called a partially biodegradable biocomposite. When matrix and reinforcement are biodegradable, they are called fully biodegradable biocomposites [13].

Biocomposites offer various advantages due to the renewable origin of their components. For example, they help to reduce dependence on petroleum-based products, promote the use of renewable raw materials, and contribute to the recovery of by-products and agricultural waste [2,14,15,16,17,18]. Biocomposites are a safe and environmentally sustainable alternative to fossil-derived materials [19,20]. Moreover, the biodegradability of these materials aligns with multiple regulations concerning plastic usage and the shift towards a circular economy across various regions, including the Americas, the Caribbean, the European Community, and the United Nations [21,22,23,24]. Due to their versatility, biocomposite materials emerge in diverse industry applications, for example, vascular prostheses [25,26], laminated materials [27], and packaging materials [28,29], among other products.

Because of their structural diversity and biodegradability, polyhydroxyalkanoates (PHAs) are exceptional for replacing plastics. PHAs are polymers produced by Gram-positive and Gram-negative bacteria from at least 75 genera [30]. The most known member of the PHA family is polyhydroxybutyrate (PHB), a highly crystalline, naturally occurring biopolymer [31,32]. PHB deficiencies in mechanical properties can be overcome by producing composites with fibers. Natural fibers used as fillers in composites improve characteristics such as stiffness, tensile strength, density, biodegradability, and the cost of the final applications [33]. Bio-sourced and biodegradable PHB composites have gained notoriety because various fibers, nanofibers, and modified fibers have been incorporated as PHB fillers, widening the range of applications from packaging to 3D printing [34,35]. Yet, the development of biocomposites requires speeding up the design process, reducing production costs, and facilitating the characterization of materials.

Materials simulation methods provide an alternative to the whole process and offer additional advantages, such as non-destructive tests, to characterize the material that significantly reduces the costs [36,37,38], optimizes models through parameterized programming [39,40], and offer the possibility of validating the methodology by comparing results with other methods and experimental data or exploring complex scenarios [41,42]. In addition, including different parameters in the simulation methodology, such as volumetric fraction, reinforcement geometries, and orientation, reduces errors due to the overestimation of mechanical properties that occur with some classical theoretical models [43,44]. With a validated model and a parametric analysis, it is possible to gain insight into the influence of different parameters and their interactions on the mechanical behavior of composites [45].

The inclusion theory, as formulated in Eshelby’s work [46], has been widely used to predict composite materials’ overall properties. This approach commonly works with a single inclusion within a sparsely distributed framework in two-phase composites [47]. While versatile, the model assumes ellipsoidal inclusions, a representation that may not cover all practical scenarios. It is essential to highlight that the Eshelby method is particularly effective in significantly diluted concentration conditions. However, various alternative methods have been devised, including the Mori–Tanaka approach. This method computes the average internal stress in the matrix of a material with uniformly distributed macroscopic inclusions. The research reveals that the average stress in the matrix remains constant throughout the material, regardless of its position within the domain [48]. Additionally, this model has been subject to refinements, which redefined and extended the Mori–Tanaka approach to apply to composite materials, accounting for anisotropic and ellipsoidal phases [49].

Another noteworthy model is the self-consistent method, based on the premise that a particle is embedded in an effective medium with unknown properties [50]. This method calculates the effective properties by iteratively adjusting the local properties of each phase until reaching self-consistency. Self-consistency occurs when the average strain in each phase, weighted by the volume fraction of that phase, equals the macroscopic strain. In contrast to the Eshelby and Mori–Tanaka methods, the self-consistent method yields implicit equations for the effective properties, often requiring numerical iteration.

The effective properties of composite materials can also be calculated using a homogenization procedure such as asymptotic homogenization (AH). The AH scheme is widely used to predict effective material properties due to its rigorous mathematical background based on perturbation theory. The boundary value problems of partial differential equations have been studied for application in many diverse engineering fields. The AH method is mathematically rigorous, but it requires a complex and lengthy derivation procedure to obtain effective material properties [51]. AH assumes that any field quantity, such as the displacement, can be described as an asymptotic expansion, which, once replaced in the governing equations of equilibrium, allows the effective properties of the composite material to be derived.

Another approach is numerical homogenization (NHM), which employs the finite element method (FEM). This numerical method demonstrates high versatility, adapting well to composites with various geometric inclusion configurations, especially those with intricate fiber arrangements [52]. NHM enables efficient calculations of effective properties by utilizing a unit cell model with appropriate periodic boundary conditions. Additionally, NMH goes beyond determining mean-field solutions for highly heterogeneous problems. It also includes the assessment of local fluctuations, a crucial consideration in numerous applications [53]. The finite element analysis obtains the local solutions and averages the effective properties of the composite [38,54,55,56], while statistical tools, such as experimental designs, reduce the number of runs (simulations) required to obtain reliable results [36,45].

This work presents the simulation, using a numerical homogenization technique based on FEM, to analyze the mechanical behavior of a composite material of a polyhydroxyalkanoate (PHA) biodegradable matrix reinforced with cylindrical inclusions of flax. This simulation considers three aspects: (1) An appropriate representation of the geometry, microstructure, and periodicity; for this purpose, a representative volume element (RVE) was used [38,56,57]; (2) Mesh element selection, which establishes adequate boundary conditions that represent the forces that act in the system [58,59,60]; and (3) An efficient numerical method to speed up the calculation time of the simulation [37,61]. The results were validated by response surface methodology, AHM, and experimental data. The simulations ran under a fractional experimental design. 

## 2. Materials and Methods

### 2.1. Experimental Design

A fractional factorial design (DDF) with three factors and tree levels was used (RStudio 2023.06.2 Build 561, Posit Software, PBC, Boston, MA, USA). The factors were chosen from the literature due to their impact on the mechanical behavior of composites: volumetric fraction ($V_{f})$, radius, and aspect ratio between the longitude and radius (ρ) of the reinforcements [62,63,64]. The factors and levels are summarized in Table 1. The selected matrix material was polyhydroxybutyrate (PHB), and the reinforcement material was flax fibers.

### 2.2. RVE Generation and Reinforcement Randomization

The RVE and the reinforcement sizing were established within the parameterization section using a random angle vector for orientation. These angles were then used to define the location of the inclusions using rotation matrices. One thousand points on the circumferences of the cylindrical reinforcements were located and validated within the RVE limits. The cell with the inclusions was generated(ANSYS APDL 2022 R1 Build 22.1 Canonsburg, PA, USA), as shown in Figure 1.

### 2.3. Meshing

The discretization of the continuous system was carried out ((SOLID 187 element ANSYS APDL 2022 R1 Build 22.1 Canonsburg, PA, USA), a tetrahedron with ten nodes and three degrees of freedom at each node. The number of elements generated depended on the number and length of reinforcements. Figure 2 displays two different RVEs with variations in fiber dimensions.

### 2.4. Boundary Conditions

Six local problems were analyzed, three for tensile stress and three for shear stress [65,66,67], summarized in Table 2. Longitudinal tension stress conditions (problem 1) were as follows: A known perpendicular load was applied to the $x=A_{x}$ face of the RVE. Likewise, the displacement of the cell in the three planes of symmetry, x=0, y=0, and z=0, was restricted to simulate the continuity of the material [55,68]. A homogeneous displacement of the nodes was ensured on the faces $x=A_{y}$, and $z=A_{z}\;$(faces in which no force was applied). The equivalent procedure was performed for the second and third scenarios.

Conditions of shear force problems (problem 4) were as follows: Two known forces were applied, equal and parallel to the x and y axes, on the faces $x=A_{x\;}$and $y=A_{y}$. The displacement in the plane of symmetry z=0 was restricted. Likewise, the movement of the face y=0 in the direction of *x* was limited, and that of the face x=0 in the y direction, to generate antisymmetric to the applied forces. Finally, a homogeneous displacement of the nodes on the $z=A_{z}$ faces was ensured. The equivalent procedure was performed for the fifth and sixth scenarios. 

### 2.5. RVE Analysis and Solution

Hooke’s law (1) was used to integrate the element’s nodes, material properties (stiffness), and deformation after applying the loads.
(1)σ=Cε,
where σ is the stress vector, ε is the strain vector, and C is the stiffness matrix. The matrix C was inverted to obtain the matrix S “compliance.” This matrix was used to rewrite the deformations as an unknown dependent change, allowing the deformation to be rewritten in terms of the stress and the elements of the matrix S [66]. The engineering constants were calculated with Equation (2).
(2)$$\left[\begin{array}{c} \varepsilon_{11}\\ \varepsilon_{22}\\ \varepsilon_{33}\\ \varepsilon_{23}\\ \varepsilon_{13}\\ \varepsilon_{12} \end{array}\right]=\left[\begin{array}{cccccc} S_{11} & S_{12} & S_{13} & 0 & 0 & 0\\ S_{21} & S_{22} & S_{23} & 0 & 0 & 0\\ S_{31} & S_{32} & S_{33} & 0 & 0 & 0\\ 0 & 0 & 0 & S_{44} & 0 & 0\\ 0 & 0 & 0 & 0 & S_{55} & 0\\ 0 & 0 & 0 & 0 & 0 & S_{66} \end{array}\right]\left[\begin{array}{c} \sigma_{11}\\ \sigma_{22}\\ \sigma_{33}\\ \sigma_{23}\\ \sigma_{13}\\ \sigma_{12} \end{array}\right].\;$$

A numerical homogenization, which relies on finite element simulations utilizing subroutines that calculate volume, deformation, and stress for each RVE element, was subsequently used to compute the engineering constants, as described in Table 3 [55].

## 3. Results

### 3.1. Validation, NHM vs. AHM

The simulation outcomes were compared with the results from the asymptotic homogenization method (AHM) [41,55]. Figure 3 shows NHM (blue asterisk) vs. AHM (green squares), revealing that the calculated engineering constants align closely. The mean average discrepancy between the two simulations was 0.0019%. The same behavior was observed with the other constants described in Table 3.

### 3.2. Simulation vs. Experimental Results

To compare the simulation results with experimental data, we used the data generated by Yan et al. from a biodegradable matrix and cylindrical inclusions to match the parameters included in our research (Table 4) [43]. The referred study used transversely isotropic fibers; therefore, it was necessary to rotate the tensioner according to the fibers’ orientation. 

Table 5 shows the NHM simulation results for two ρ and the error estimations. Because the ρ  was not specified, two approximate data were used, taking the Yan et al. micrographs as a reference. The ρ  dramatically impacted the results, especially the longitudinal module, probably due to the inclination angles used for the fibers’ rotation.

### 3.3. Mechanical Behavior of the Simulated Composite

Figure 4a,b showed that an increase in ρ or $V_{f\;}$ results in an increment in the modulus $E_{11}$. Additionally, it is possible to identify an interaction between these factors. When ρ =1, the mean value of $E_{11}$ is 3686 MPa for $V_{f\;}$ = 8% and 3790 MPa was the mean value for $V_{f\;}$ = 12%, representing an approximate 2.7% increase. However, for ρ = 3, the value of $E_{11}$ is 4150 MPa for $V_{f\;}$ = 8%, and 4555 MPa for $V_{f\;}$ = 12%, indicating a 9% increase in the modulus. This suggests that as ρ increases, the impact of $V_{f\;}$ on reinforcing the elastic modulus of composite becomes more evident.

A similar effect is observed in the case of the modulus $G_{12}$. Figure 4c,d, for ρ = 1, the change in $V_{f\;}$ from 8% to 12% appears to have an insignificant impact on the stiffness of the composite. However, when ρ = 3, the increase in $V_{f\;}$ results in a 7% stiffness increment. Regarding the coefficient $v_{12}$. In Figure 4e,f, an increase in ρ leads to an approximately 1% decrease in $v_{12}$, which aligns with the increased composite’s stiffness. In the case of $v_{12}$, the interaction effect of the factors is less pronounced, but there is greater variability in the data. The variability of the factor can be examined through the box plot presented in Appendix A.

### 3.4. Linear Model and Contour Plot to Predict Elastic Modulus

The statistical analysis of the simulation results appears in Figure 5. A Pareto diagram (Figure 5a) highlighted the influence of the Young’s Modulus factor. Among these factors, ρ and $V_{f\;}$were the most significant contributors, and notably, their interaction also had a positive effect on the elastic modulus. In contrast, the Ratio(E_fiber/E_matrix) and the radius size exhibited comparatively less impact on Young’s Modulus.

The prediction model for Young’s Modulus is expressed by Equation (3) and excludes the radius factor due to its little impact on the response variable.
(3)E11=4004.16+304.75ρ+127.75Vf+26.9β+40.06ρ^2+76.54ρ  Vf+16.91ρ β+7.43Vf β+5.04 β Vf ρ
where: Young’s Modulus $\left(E_{11}\right)\;;$  Volumetric Fraction $\left(V_{f}\right)$; Aspect Ratio (ρ); Modulus Ratio (β).

Figure 5b reveals insightful interactions between the factors. This graph used $V_{f}\;$ and ρ due to significant influences. The $V_{f}$-axis indicates that an increment in $V_{f}\;$had a slight effect on Young’s Modulus, but as ρ increased, the influence of the $V_{f}\;$ became more pronounced. The response surface pointed to the optimal approach for enhancing the elastic modulus. The color map (blue–yellow) indicates the direction of improvement. 

## 4. Discussion

In this work, we modeled the mechanical behavior of a composite material of a polyhydroxyalkanoate (PHA) biodegradable matrix reinforced with cylindrical inclusions of flax using the NHM. We validated our results with the AHM method for its accurate prediction of macroscopic behaviors [69,70]. This characteristic is pivotal, especially when comparing simulation outcomes. AHM has proven helpful in the simulation work of short fiber composites compared with NHM in three-dimensional cell cases [71]. 

For a second validation, we compared our NHM results with published experimental data, with errors ranging from 0.33% to 6.45% obtained for the transverse and longitudinal moduli, with a noticeable increase in error as ρ increased. Several aspects could contribute to this discrepancy. First, the simulations assumed perfect contact between the matrix and reinforcement, which may not hold in experimental conditions [72,73,74]. Second, it was assumed that the reinforcements had uniform geometry, a simplification that does not align with real-world variations. Finally, the simulated material did not account for aggregation. All these aspects could introduce additional rigidity to the composite material. Future research will enhance the algorithm by incorporating a probability distribution to simulate agglomerations and geometric variations. Nonetheless, the error percentage falls within the range reported by other algorithms for predicting mechanical properties. Some studies suggest that error prediction is due to non-linear elastic behavior or estimation errors when the mesh size is too small. Despite the limitations of the present model, we considered it worthwhile to obtain a good approximation of the compound and to reduce experimental costs [36,71,75,76,77,78,79].

The significance of the ρ  of reinforcement was underscored in our statistical analysis, a finding that aligns with various experimental studies. For instance, in the case of a PLC–hemp mixture, it was observed that Young’s Modulus increased by 48% as the ρ of the reinforcements changed from 19 to 26. However, when this ratio exceeded 30, a saturation effect occurred, likely attributable to moisture absorption by the fibers [80]. Additionally, another study [81] supports the notion that particles with larger aspect ratios are more effective in enhancing composite stiffness. In their study, a ρ of 10, combined with a volume fraction of 30%, resulted in a 4.5% increase in Young’s Modulus and a 1.3% decrease in Poisson’s coefficient. Nevertheless, they also observed a saturation effect when the ρ exceeded 10, suggesting the occurrence of a “shear lag” phenomenon that limits stress transmission from the matrix to the fibers. In a PHBV–potato mixture, researchers have noted that fibers with small aspect ratios tend to function more as fillers than reinforcements [82]. 

When the matrix is reinforced with long particles or fibers and the fiber length is lower than the length of RVE, the matrix and fiber interact as springs connected in series, resulting in a high reinforcement effect. However, if the fiber length exceeds the dimensions of the RVE, the interactions resemble parallel springs, leading to a diminished reinforcement effect. The *β* and the ρ  also interact, and when *β* is >10, these interactions become dominating factors in determining the directional effective Young’s Modulus [83].

Of interest, Robinson et al. [84] highlighted that the effectiveness of fiber length in reinforcing a material depends not only on its diameter but also on the ratio between the elastic modulus of the reinforcement and the resin (β), the Poisson coefficient, and the volume fraction. This relationship is mathematically presented in Equation (4).
(4)$$l_{c}=2.3\;d\sqrt{\frac{E_{f}\;\left(1+v_{m}\right)}{E_{m}}}\;\sqrt{ln{\left(\frac{\pi }{4V_{f}}\right)}^{1/2},\;}$$
where: The fiber diameter (*d*), the modulus fiber (*E_f_*), the matrix modulus (*E_m_*), the matrix poisson (*v_m_*), the volumetric fraction (*V_f_*). According to this relationship, the minimum length of the reinforcements should be 6 mm; however, the longest inclusion used in the simulation is 2.8 mm, because of which the elastic modulus of the matrix increased between 17 and 31% and the shear by 15–24%. The effect of the β was also analyzed, but for the simulated cases, the effect of the β is not so significant.

Additionally, most studies concur on the influence of ρ on the Elastic Modulus, but there are notable exceptions. In the epoxy–nanocarbon fibers research, the authors reported that reinforcements with a lower ρ  led to a greater apparent Young’s Modulus [85], attributed to enhanced interfacial adhesion. Also, the authors reported that reinforcements with a lower ρ  led to a greater apparent Young’s Modulus. This phenomenon results from enhanced interfacial adhesion due to better dispersion and alignment of short fibers. The researchers also emphasized that the$\;V_{f}\;$had a more significant impact than the ρ opposite to the simulation results [85]. Furthermore, another study analyzing a PHBV–cellulose composite [86] showed that weak interfacial adhesion softened the matrix modulus, particularly when employing reinforcements with a small ρ.

The simulation works have also analyzed the reinforcement orientation; unidirectional fibers increase longitudinal modulus by 26%, but the transverse modulus decreases to 4.75. This effect depends on interacting with the $V_{f}$ [87]. Other work states that a simulation with random fibers is closer to the experimental data [88]. The proposed methodology uses random reinforcement orientation to simulate the injected composite material. In the experimental validation, the fiber rotation was limited to −30° to 30° since the micrographs of the work used as a reference showed limited fiber rotation [43].

The last methodology step presented in this research proposed a non-linear regression model, Equation (3), to predict the Young’s Modulus of the compound. The model is considered helpful in predicting the mechanical properties of the composite when parameters such as ρ and $V_{f}$ are varied. For example, increasing ρ to 9.5 and $V_{f}$ to 20% (in coding units) and then substituting into Equation (3) while keeping the β constant estimates the Young’s Modulus ($E_{11}$) to be 7515.2 MPa.

It is necessary to consider that the model is a generalization and may incorporate noise. Additionally, the model assumes a constant relationship between the factor and$\;E_{11}$, which can lead to inaccurate predictions when the factor’s level is changed to values significantly distant from those simulated. Non-linear regression models are sensitive to the initial parameter values.

Although this simulation methodology might seem simple, it allows precise predictions of the mechanical behavior of composites, including changes in elastic and shear modulus, in scenarios involving fibers with approximate cylindrical shapes. The regression model enhances result reliability, leading to reduced computational costs, design, and production times for materials with similar characteristics (*β*,ρ,*V_f_*). For example, it is possible to utilize the model to predict the elastic modulus of a PHB-Jute composite with *V_f_* = 16%, *β* = 7.7, and ρ = 5, resulting in *E*_11_ = 4800 MPa (approximately). Worth noting is that the structure of the composites is complex and requires the design of complex algorithms that include random distributions of the reinforcements in the unit cells, include movement restrictions of the nodes and planes, and thus represent the reality of a composite material.

## 5. Conclusions

Many fibers are industrial waste and are underused. Biodegradable matrix composites with fiber reinforcements have been developed because they could potentially replace polymers in olefins. Composites of PHB with natural fibers have great potential for industrial applications. The algorithm developed here facilitated the simulation of the reinforcement of a PHB matrix with cylindrical fiber inclusions and produced reliable results. The algorithm’s parameters helped simulate different configurations of the materials. This algorithm relied on the geometric depiction of the material and used numerical homogenization methods for analysis. This model is a valuable tool for predicting both longitudinal and transverse elastic moduli, shear modulus, and Poisson’s coefficient. Furthermore, a linear regression model was developed, simplifying the prediction of mechanical properties for alternative material configurations while mitigating the need for extensive computational resources. This simulation methodology sets the basis for future work, including challenging conditions such as porous cells, aggregations, and inclusions with different aspect ratios. This advancement aims to enhance the representation of composite materials in a simulation process. 

## Figures and Tables

**Figure 1 polymers-15-04727-f001:**
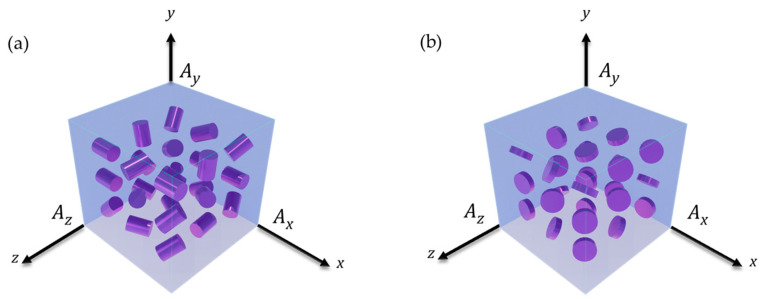
RVEs with randomly oriented cylindrical inclusions, (**a**) ρ = 5; (**b**) ρ = 1.

**Figure 2 polymers-15-04727-f002:**
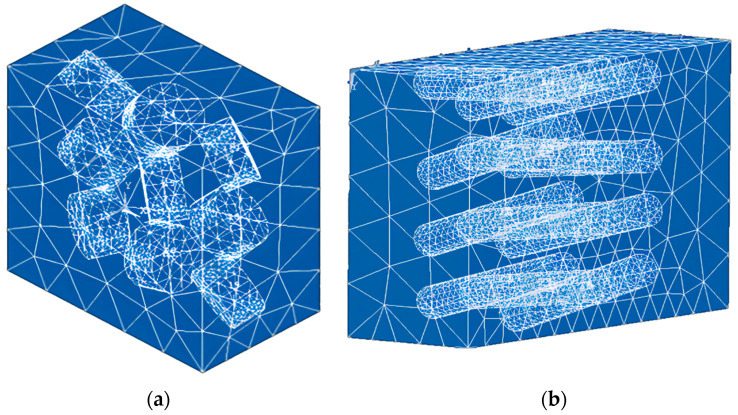
The meshing of RVEs using SOLID 187 element: (**a**) ρ = 1 with nine inclusions; (**b**) ρ = 10 with twelve inclusions.

**Figure 3 polymers-15-04727-f003:**
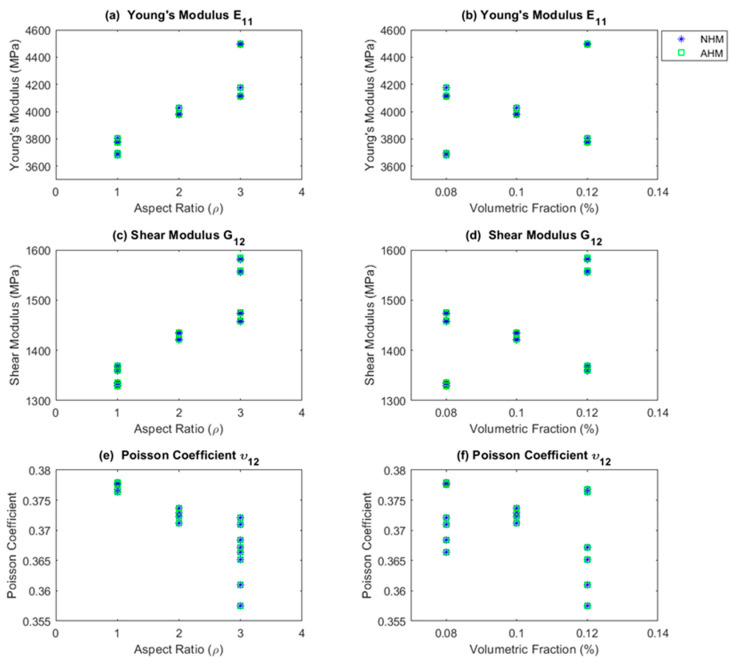
Comparison between NHM and AHM. In (**a**,**c**,**e**), the *x*-axis represents the ρ. In (**b**,**d**,**f**), the *x*-axis represents the percentage of reinforcements included in the matrix, $V_{f}$.

**Figure 4 polymers-15-04727-f004:**
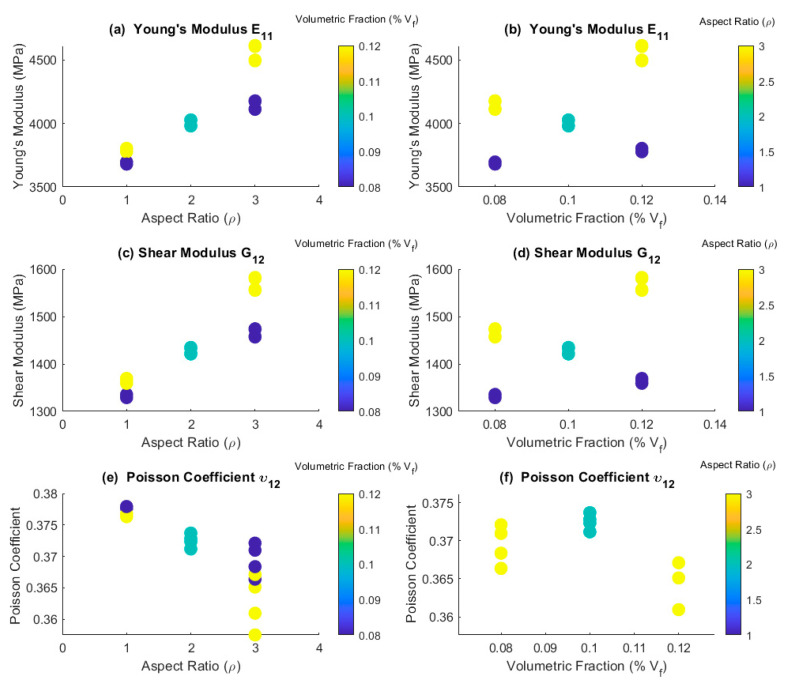
Comparison between NHM and AHM. In (**a**,**c**,**e**), the *x*-axis represents the ρ, and the color bar represents $V_{f}$. In (**b**,**d**,**f**), the *x*-axis represents the percentage of reinforcements included in the matrix $V_{f}$ and the color bar represents ρ.

**Figure 5 polymers-15-04727-f005:**
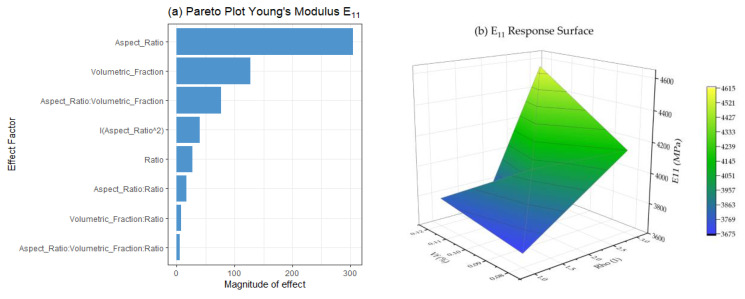
(**a**) $E_{11}$ Pareto plot showing the positive impact of the analyzed factors on elastic modulus; (**b**) $E_{11}$ response surface.

**Table 1 polymers-15-04727-t001:** Fractional factorial design, three factors, and three levels.

Factor	Low Level	Unit Code	Medium Level	Unit Code	High Level	Unit Code
Radius	0.0625	−1	0.125	0	0.1875	1
ρ	1	−1	3	0	5	1
$V_{f}$	8%	−1	12%	0	16%	1

**Table 2 polymers-15-04727-t002:** Boundary conditions for tensile and shear stress.

Local Problem	Illustration	Symmetric Mirror Face	Anti Symmetry Face	Load Face	Constant Displacement Faces
Problem 1: Tensile stress. Axis = x	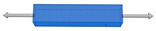	x=0, y=0, z=0.	none	x=Ax	y=Ay, z=Az.
Problem 2: Tensile stress. Axis = y	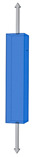	x=0, y=0, z=0.	none	y=Ay	x=Ax, z=Az.
Problem 3: Tensile stress. Axis = z	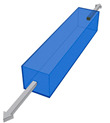	x=0, y=0, z=0.	none	z=Az	x=Ax, y=Ay.
Problem 4: Shear stress plane = xy	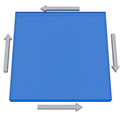	z=0.	y=0, restrict movement in direction x.	x=Ax, y=Ay	z=Az.
x=0, restrict movement in direction y.
Problem 5: Shear stress plane = y*z*	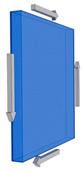	x=0.	y=0, restrict movement in direction *z*.	$y=A_{y}$ $z=A_{z}.$	$x=A_{x}.$
z=0, restrict movement in direction y.
Problem 5: Shear stress plane = x*z*	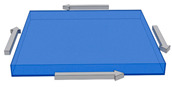	y=0.	*x* = 0, restrict movement in direction *z*	$x=A_{x}$, $z=A_{z}.$	$y=A_{y.}$
*z* = 0, restrict movement in direction *x*

**Table 3 polymers-15-04727-t003:** Compliance element matrix and engineering constants for orthotropic materials [66].

Problem	Element Matrix S	Engienering Constat	
		Constant	Definition
1	$s_{11}=\frac{1}{E_{11}}$	$E_{11}$	$\;E_{11}=\frac{{\bar{\sigma }}_{11}}{{\bar{\varepsilon }}_{11}}$
	$s_{12}=\frac{-v_{12}}{E_{22}}$	$v_{21}$	$v_{21}=-\frac{{\bar{\varepsilon }}_{22}}{{\bar{\varepsilon }}_{11}}$
	$s_{13}=\frac{-v_{32}}{E_{33}}$	$v_{31}$	$v_{31}=-\frac{{\bar{\varepsilon }}_{33}}{{\bar{\varepsilon }}_{11}}$
2	$s_{21}=\frac{-v_{12}}{E_{11}}$	$v_{12}$	$v_{12}=-\frac{{\bar{\varepsilon }}_{22}}{{\bar{\varepsilon }}_{11}}$
	$s_{22}=\frac{1}{E_{22}}$	*E* _22_	$E_{22}=\frac{{\bar{\sigma }}_{22}}{{\bar{\varepsilon }}_{22}}$
	$s_{23}=\frac{-v_{32}}{E_{22}}$	$v_{32}$	$v_{32}=-\frac{{\bar{\varepsilon }}_{22}}{{\bar{\varepsilon }}_{33}}$
3	$s_{31}=\frac{-v_{13}}{E_{11}}$	$v_{12}$	$v_{32}=-\frac{{\bar{\varepsilon }}_{22}}{{\bar{\varepsilon }}_{33}}$
	$s_{32}=\frac{-v_{23}}{E_{22}}$	$v_{23}$	$v_{32}=-\frac{{\bar{\varepsilon }}_{33}}{{\bar{\varepsilon }}_{22}}$
	$s_{33}=\frac{1}{E_{33}}$	$\;E_{33}$	$\;E_{33}=\frac{{\bar{\sigma }}_{33}}{{\bar{\varepsilon }}_{11}}$
4	$S_{44}=\frac{1}{G_{23}}$	$G_{23}$	$G_{23}=-\frac{{\bar{\sigma }}_{11}}{2\left({\bar{\varepsilon }}_{22}+{\bar{\varepsilon }}_{33}\right)}$
5	$S_{55}=\frac{1}{G_{13}}$	$G_{12}$	$G_{13}=-\frac{{\bar{\sigma }}_{11}}{2\left({\bar{\varepsilon }}_{11}+{\bar{\varepsilon }}_{33}\right)}$
6	$S_{66}=\frac{1}{G_{23}}$	$G_{13}$	$G_{12}=-\frac{{\bar{\sigma }}_{11}}{2\left({\bar{\varepsilon }}_{11}+{\bar{\varepsilon }}_{22}\right)}$

**Table 4 polymers-15-04727-t004:** Experimental data were used for simulation.

	Matrix (PLA)	Reinforcements (Carbon Fibers)
Young’s Modulus ($E_{11}$)	2570	207,000
Young’s Modulus ($E_{22}$)		14,000
$\mathrm{Poisson}\;\mathrm{Coefficient}\;(v_{23}$)	0.3	0.25
$V_{f}$	0.85	0.15
Average length		77.1 µm

**Table 5 polymers-15-04727-t005:** Experimental vs. simulation results by NHM.

Engineering Constant	Experimental Data(MPa)	NHM (MPa) ρ=10	%Error	NHM (MPa) ρ=13	%Error
Young’s Mogulus ($E_{11}$)	5030	5012.9	0.33	5961.5	7.8
Young’s Modulus $(E_{22}$)	3720	3480.6	6.45	3416.7	7.45

## Data Availability

Data are contained within the article.

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
