# Peer review of "Simulation of a Composite with a Polyhydroxybutyrate (PHB) Matrix Reinforced with Cylindrical Inclusions: Prediction of Mechanical Properties"

_polymers, 2023, doi:10.3390/polym15244727_

Round 1
Reviewer 1 Report
Comments and Suggestions for Authors
The authors performed good work on “Simulation of a composite with a biodegradable matrix reinforced with randomly oriented cylindrical inclusions: numeric analysis, validation, and prediction of mechanical 4 properties:
It is acceptable while considering/adding the following observations:
1. Mention the exact/ specific material name in the title and abstract for which this study is being carried out.
2. Mention the how the constituents’ fractions effect the composite material.
3. Conclusions section should be revisited with novelty of this research.
4. English grammar and proof reading is recommended to check and improve.
Comments on the Quality of English LanguageLanguage and grammar needs to be improved
Author Response
Comments and Suggestions for Authors
The authors performed good work on “Simulation of a composite with a biodegradable matrix reinforced with randomly oriented cylindrical inclusions: numeric analysis, validation, and prediction of mechanical 4 properties:
It is acceptable while considering/adding the following observations:
Point 1. Mention the exact/ specific material name in the title and abstract for which this study is being carried out.
Response 1: Thanks for the suggestion; we have considered it and included the specific material name in the title and abstract.
Point 2. Mention the how the constituents’ fractions effect the composite material.
Response 2: In lines 204 -231, we expanded the information on how the constituent's fraction affects the mechanical properties, and we also improved Figure 4 to reinforce this information.
Point 3. Conclusions section should be revisited with novelty of this research.
Response 3: We have addressed the suggestion, and with the deepening of the conclusions, we have improved the perspective of the document.
Point 4. English grammar and proof reading are recommended to check and improve.
Response 4: We ran a grammar and spelling check using specialized software.
Reviewer 2 Report
Comments and Suggestions for Authors
Horacio Vieyra et al., in their manuscript entitled "Simulation of a composite with a biodegradable matrix reinforced with randomly oriented cylindrical inclusions: numeric analysis, validation, and prediction of mechanical properties", prepared a model for predicting longitudinal and transverse elastic moduli, shear modulus, and Poisson's coefficient using the Representative Volume Element and Numerical Homogenization Method. However, some of the deficiencies listed below were observed:
1. The manuscript contains some typos and spelling mistakes. They should be corrected.
2. In the introduction, there is not any information about similar published models and their pros and cons. The authors should add this information. However, the authors have already used a very large number of references (about 60) in the introduction, which may be more useful to improve the introduction.
3. Could the authors add an explanation of why they chose polyhydroxybutyrate (PHB) as a matrix material and flax fibers as a reinforcement material?
4. The proposed model is too simple, and there is no discussion about the changes in the structure of polymer/filler composites and how that will affect the mechanical properties. Why will this model be important to the readers?
5. In conclusion, the authors should add some perspectives to this work.
Comments on the Quality of English LanguageModerate editing of the English language is required.
Author Response
Comments and Suggestions for Authors
Horacio Vieyra et al., in their manuscript entitled "Simulation of a composite with a biodegradable matrix reinforced with randomly oriented cylindrical inclusions: numeric analysis, validation, and prediction of mechanical properties", prepared a model for predicting longitudinal and transverse elastic moduli, shear modulus, and Poisson's coefficient using the Representative Volume Element and Numerical Homogenization Method. However, some of the deficiencies listed below were observed:
Point 1. The manuscript contains some typos and spelling mistakes. They should be corrected.
Response 1: We really appreciate the observation. The typos and spelling mistakes were corrected.
Point 2. In the introduction, there is not any information about similar published models and their pros and cons. The authors should add this information. However, the authors have already used a very large number of references (about 60) in the introduction, which may be more useful to improve the introduction.
Response 2: In lines 72-110, we added information about similar published models, as suggested. We have also improved the introduction by organizing the information and taking advantage of the references.
Point 3. Could the authors add an explanation of why they chose polyhydroxybutyrate (PHB) as a matrix material and flax fibers as a reinforcement material?
Response 3: In lines 51-60, in the introduction section, we added information explaining the importance of PHB as a matrix material and flax fibers as a reinforcement material in a simulation process.
Point 4. The proposed model is too simple, and there is no discussion about the changes in the structure of polymer/filler composites and how that will affect the mechanical properties. Why will this model be important to the readers?
Response 4. In lines 204 -231, we have expanded the information on how the structure of a composite material affects the mechanical properties, and we also improved Figure 4. In lines 287-292, we also included a paragraph to explain the relationship between geometry and its effects on the reinforcement. Regarding the simplicity of the model, in lines 333-343, we added information about the extent and the importance of a good algorithm design that correctly represents the composite material.
Point 5. In conclusion, the authors should add some perspectives to this work.
Response 5. We appreciate the suggestion. With the reviewer’s concerns addressed and the improvement of the conclusions, we have broadened the perspective of the document.

Round 2
Reviewer 2 Report
Comments and Suggestions for Authors
The author made sufficient improvements to the manuscript. However, two of the deficiencies listed below were observed:
- The title of the manuscript is too long; the author should reformulate it.
- The resolution of all figures should be improved.
Comments on the Quality of English Language
Minor editing of the English language is required.
Author Response
Rewiewer 2.
Round 2.
The author made sufficient improvements to the manuscript. However, two of the deficiencies listed below were observed:
Point 1. The title of the manuscript is too long; the author should reformulate it.
Response 1: Thanks for the suggestion. We have reformulated the title of the document and it is shorter.
Point 2. The resolution of all figures should be improved.
Response 2: The figures have been modified and now the figures have higher resolution, as suggested.
